# Effect of Fractionation and Processing Conditions on the Digestibility of Plant Proteins as Food Ingredients

**DOI:** 10.3390/foods11060870

**Published:** 2022-03-18

**Authors:** Andrea Rivera del Rio, Remko M. Boom, Anja E. M. Janssen

**Affiliations:** Food Process Engineering, Wageningen University, 6700 AA Wageningen, The Netherlands; andrea.riveradelrio@wur.nl (A.R.d.R.); remko.boom@wur.nl (R.M.B.)

**Keywords:** plant protein digestibility, protein isolates, protein concentrates, alternative fractionation, heat treatment, protein modifications, in vitro protein digestion

## Abstract

Plant protein concentrates and isolates are used to produce alternatives to meat, dairy and eggs. Fractionation of ingredients and subsequent processing into food products modify the techno-functional and nutritional properties of proteins. The differences in composition and structure of plant proteins, in addition to the wide range of processing steps and conditions, can have ambivalent effects on protein digestibility. The objective of this review is to assess the current knowledge on the effect of processing of plant protein-rich ingredients on their digestibility. We obtained data on various fractionation conditions and processing after fractionation, including enzymatic hydrolysis, alkaline treatment, heating, high pressure, fermentation, complexation, extrusion, gelation, as well as oxidation and interactions with starch or fibre. We provide an overview of the effect of some processing steps for protein-rich ingredients from different crops, such as soybean, yellow pea, and lentil, among others. Some studies explored the effect of processing on the presence of antinutritional factors. A certain degree, and type, of processing can improve protein digestibility, while more extensive processing can be detrimental. We argue that processing, protein bioavailability and the digestibility of plant-based foods must be addressed in combination to truly improve the sustainability of the current food system.

## 1. Introduction

The current food production system is not sustainable [1]. The largest environmental impact can be attributed to the production of animal-based protein [2]. One of the measures proposed by Willett et al. [3] to reduce this negative impact is to lower our consumption of foods of animal origin and to increase that of plant-based foods. To facilitate this transition, plant-based alternatives to meat, dairy and eggs are continuously introduced in the market. With the increase in flexitarian diets, there is a growing demand in the vegetarian and vegan food space [4]. Nevertheless, the extensive transformation and purification of the ingredients, in addition to the lower protein yield from crop to food product, limits the sustainability potential [5,6].

Generally, plant proteins present less favourable techno-functional properties compared to milk proteins, particularly those depending on solubility such as gelling, emulsifying and foaming properties [7]. In addition, it is not clear yet whether plant- and animal-based proteins can be interchangeable from a nutritional point of view. The dietary requirement of indispensable amino acids (AA) can be satisfied by proteins from various crops [8,9]. Antinutritional factors, digestibility and bioavailability must also be considered when assessing the nutritional quality of proteins. The in vivo protein digestibility-corrected amino acid score (PDCAAS) [10] and in vitro digestibility [11] of some protein-rich ingredients and whole foods have been reported. Furthermore, the effect of domestic and industrial processing on digestion of proteins from legumes consumed as a whole food or flour, i.e., not as a protein-rich ingredient, has been reviewed [12].

Plant proteins are diverse, and most constitute a mixture of various protein units, each with its own properties. For instance, varieties of the same legume species have different globulin to albumin ratios. Globulins have been found to be more susceptible to hydrolysis by digestive enzymes [13,14]. Moreover, 7S and 11S globulin-rich protein fractions from hemp protein isolate (PI) presented different in vitro digestion profiles [15]. Yang et al. [16] found that higher proportions of β-7S subunits had a detrimental effect on the in vitro digestibility of soybean PI. Protein concentrates (PC) from different cultivars of the same species can present different structural, thermal, techno-functional properties and nutritional value, such as the indispensable AA content and digestibility, as was found for rice and millet proteins [17,18]. Meanwhile, different varieties of lupin and sorghum differ in composition and structural properties but are digested to a similar extent [19,20]. This already suggests that the digestibility of proteins from different plant sources might not be affected in the same way by a given type of processing.

The objective of this article is to review the large body of data on the digestion of protein-rich ingredients and on how processing, before, during or after the extraction of the ingredient, may alter it. We recognize the breadth of protocols used to simulate digestion as well as the methods used to describe or quantify the extent of it (Figure 1). As these confounding factors contribute to variations in results, we limited this review to studies that compare some treatment or processing to a control and noted the effect on protein digestibility of a given ingredient.

There is some disparity in the number of studies favouring some types of processing over others, as well as some crops over others. Moreover, the wide range of digestion assays makes it relatively futile to quantitatively compare results from different studies. We therefore present a narrative review with elements of a systematic one, instead of a full systematic review with meta-analysis.

## 2. Method and Definitions

Review characteristics: The search query used in Scopus was: (“protein” W/6 digest*) AND “in vitro” AND “human” AND (“gastric” OR “intestinal” OR “gastrointestinal” OR “pepsin” OR “trypsin”). In PubMed, the MeSH terms for “plant proteins, dietary” and “digestion” were also included. From the results, the works considering some measure of digestibility or protein hydrolysis by digestive enzymes, simulating some physiological condition(s), were included. Studies on whole foods or flours were not considered, as these sources have been studied elsewhere. Articles studying feed, e.g., for ruminal digestion, emulsions, animal-sourced foods or proteins, and works dealing with allergenicity or immunoreactivity were excluded.

The term “protein digestibility” is used rather ambiguously throughout the reviewed literature. By definition, digestibility is the proportion of an ingested food or nutrient that can be absorbed into the bloodstream or body. However, it is also used to describe protein degradability, i.e., the proportion of intact protein remaining, the resulting degree of hydrolysis (DH) or the proportion of low molecular weight peptides resulting after the action of digestive enzymes. Other measurements of digestibility are listed in Figure 1 and details of the digestion assays and measurements for each of the studies reviewed are listed in the Appendix A.

Figure 2 presents a scheme of the different processes reviewed. Throughout the text, “conventional aqueous fractionation” refers to milling, optional defatting for oil-containing seeds, alkaline extraction, centrifugation, isoelectric precipitation, centrifugation, washing and freeze drying, as it is mostly performed in laboratory setting, or spray drying, more common in commercially available ingredients. Table 1 summarizes the effects on digestibility of the more commonly studied processes for different plant sources.

## 3. Ingredient Preparation

### 3.1. Pre-Fractionation Treatment

Most commonly, seeds are milled into a flour or grits prior to alkaline extraction. Soaking seeds at high temperatures, before milling for conventional aqueous fractionation, was shown to improve the in vitro digestibility of soybean and cowpea PI. In the work of Wally-Vallim et al. [21], PI from soybean seeds soaked at 40 °C was more digestible than at 60 °C. The in vitro gastric digestibility was improved by longer soaking times for both temperatures. It was argued that at 40 °C, proteins were partially denatured, while at 60 °C the 7S fraction was completely denatured, and protein structures had rearranged. Meanwhile, PI from soaked and autoclaved cowpea seeds was more extensively hydrolysed by pepsin–pancreatin than that from raw seeds [22].

Some studies explored the effect of germination prior to fractionation of soybean and black bean. A direct relation between the germination time and the extent of hydrolysis achieved by digestive enzymes was observed [23,24]. Concurrently, the trypsin inhibitory activity (TIA) was reduced by germination, associated to protease-catalysed hydrolysis of lectins and trypsin inhibitors. Aijie et al. [25] found a similar relation; however, the DH decreased, and the TIA increased for the longest germination times, which they explained by a resynthesis of trypsin inhibitors by photosynthesis. For black soybean, an inverse relation was observed: the PI produced from non-germinated seeds yielded the largest proportion of low molecular weight peptides [26]. It was hypothesised that these small peptides were used for tissue formation during germination. 

Solid state and submerged fermentation of milled lupin with different strains of *Pediococcus* prior to subsequent conventional aqueous fractionation improved the in vitro protein digestibility in the PI compared to the non-fermented control [27]. At the same time, the fermentation reduced the content of trypsin inhibitors. No clear relation can be drawn between the type of fermentation and digestibility, as many different lupin hybrid lines and strains of *Pediococcus* were studied. 

### 3.2. Conventional Protein Fractionation

After a defatted meal has been obtained, alkaline extraction is the first step in conventional aqueous fractionation. Higher protein purities, at the expense of lower yields, can be obtained with increasingly higher concentrations of a strong alkali, typically NaOH. Alkaline treatment has been associated with the formation of lysinoalanine and AA isomerisation in rice residue PI, reducing the in vitro digestibility and absorption in a rat model [28]. Protein extracted from defatted lupin meal at acidic pH (pH 2) was more readily and extensively digested than that extracted at neutral or alkaline pH (pH 8.5) conditions, using an in vitro digestion assay [29]. The extraction pH was thought to induce different structural conformations and extents of denaturation. Nevertheless, Ruiz et al. [30] did not find a significant effect on the in vitro gastric digestion of quinoa PI extracted at pH 8 to 11. 

Either PC or PI can be obtained from the conventional fractionation process. Commercial PC and PI have been used in in vivo rat assays, showing a small variation in PDCAAS, the true or standard ileal digestibility, of soybean ingredients [31,32]. Meanwhile, the in vitro gastric digestibility of commercial soybean PI remained unchanged after long-term storage at freezing and high temperatures [33].

### 3.3. Alternative Protein Fractionation Strategies

Modifications to the conventional aqueous fractionation process have been proposed to improve the purity, yield or techno-functional properties of the ingredients obtained. Conventionally, alkaline extraction is performed with NaOH, with the pH adjustment for isoelectric precipitation performed with HCl. Chamba et al. [34] proposed the use of alkaline ash from burnt green and purple amaranth and lemon juice as “natural” alternatives to the more commonly used chemicals to isolate soybean protein from full fat and defatted flour. The PDCAAS was slightly higher for the material extracted with “natural” chemicals, while no significant difference was observed between the in vitro pepsin–pancreatin digestibility of “natural” and conventional chemicals. The use of conventional chemicals was somewhat more effective at reducing the content of antinutritional factors such as trypsin inhibitors and phytic acids in PI. Na_2_SO_3_ has been used to extract proteins and to prevent oxidative darkening of the PI, from lupin and chickpea. The digestibility of Na_2_SO_3_-extracted lupin PI was higher than the conventionally fractionated ingredient [35]. However, for chickpea PI, the digestibility from both extractions did not differ [36]. 

Ultrafiltration has been used as an alternative to isoelectric precipitation. The TIA was similarly reduced by either process for soybean PI [37]. The extent of hydrolysis achieved with pepsin–pancreatin digestion, as well as the reduction of the TIA, was comparable for brown lentil PI separated by ultrafiltration and for conventional isoelectric precipitation [38].

The effect of different drying methods on the protein digestibility was studied for buckwheat and hempseed PI. Tang [39] showed that freeze drying, compared to spray drying, produces buckwheat PI that is better digestible by pepsin–trypsin. However, when alkaline extraction was assisted by ultrasonic treatment instead of by just mechanical stirring, freeze- and spray-dried PI were equally digestible. Meanwhile, Lin et al. [40] compared vacuum oven, oven or freeze drying of hempseed PC. In this study, freeze drying also produced better digestible PC compared to drying at higher temperatures, which was attributed to the formation of poorly digestible Maillard products during oven or vacuum oven drying.

Enzyme-assisted fractionation paired with extrusion has been presented as an environmentally friendly alternative to conventional aqueous fractionation [41]. Oil and protein were simultaneously extracted from soybean flakes that were extruded and treated with a bacterial endoprotease under alkaline conditions to obtain oil-, fibre- and protein- and sugar-rich fractions. Extrusion or enzyme action during processing did not alter the pepsin digestibility of the resulting ingredients, although some techno-functional properties were improved. Extrusion and α-amylase-catalysed starch liquefaction were used to concentrate proteins from white sorghum [42]. While the moisture content in the barrel during extrusion influenced the in vitro gastric digestibility, no effect from α-amylase action was observed. Nevertheless, the sorghum PC showed lower digestibility than sorghum flour. This was attributed to re-aggregation during the boiling step that was used for enzyme inactivation.

Air classification is a dry fractionation technique. The digestibility of pea, lentil and fava bean PC obtained from air classification were compared to that of NaCl-extracted PI from aqueous fractionation in a mice study [43]. Overall, the digestibility of the PC was lower than that of the PI, most significantly for pea. Likewise, air-classified fava bean PC was less extensively hydrolysed during pepsin–pancreatin digestion than a PI from isoelectric precipitation and spray drying [44]. Further, the TIA from the initial flour was maintained in the air-classified ingredient and significantly reduced in the conventionally produced PI. Conversely, air jet-sieved quinoa PC was slightly more extensively hydrolysed by pepsin than a conventional aqueous-fractionated PI [45]. We hypothesize that the protein denaturation achieved through heating during spray drying facilitates the access of digestive enzymes to the cleavage sites within the proteins. 

## 4. Post-Fractionation Processing

Protein ingredients are further processed into finished products. The effects of different protein steps (fermentation, ultrasound treatment, heating, protein modification, among others) have been researched on PI and PC from various crops. Ultrasonic treatment of fava bean PI dispersions slightly reduced the in vitro digestibility [46].

Fermentation of commercial pea PC with *Lactobacillus plantarum* had a positive effect on the in vitro protein digestibility and a reduction of antinutritional factors, phenols, tannins, chymotrypsin and trypsin inhibitors. Nevertheless, the in vitro PDCAAS was negatively impacted. This was explained by the catabolism of sulphur-containing AA by the lactic acid bacteria [47]. Similarly, *L. plantarum*-fermented soybean PI released more free AA than the non-fermented control, in a dynamic in vitro gastrointestinal digestion assay [48]. Additionally, protein aggregation was observed in the gastric phase only for the non-fermented PI, as well as a higher proportion of high molecular weight peptides at the beginning of the intestinal phase.

### 4.1. Proteolysis

Protein hydrolysis has mixed effects on protein digestibility. For soybean protein, hydrolysis by immobilized trypsin improved or had no effect on the extent of digestion [49]. In this study, pre-digested proteins were better digestible under infant gastric condition, simulated by a less acidic pH (pH 4) compared to adult models. Meanwhile, a soybean protein pepsin–hydrolysate was as digestible as the intact PI, in a different infant model with reduced digestive enzyme concentration, compared to an adult model [50]. 

A series of studies investigated the effect of the co-ingestion of soybean PI and dietary actinidin from green kiwifruit extract on the protein digestion. From an in vitro pepsin–pancreatin assay, some subunits such as the 11S basic polypeptide showed some effect of the actinidin; however, no overall effect on the protein degradability was observed [51]. From an in vivo rat study, the presence of actinidin in the diet showed no significant effect on the true ileal digestibility of soybean PI [52]. Gastric chyme samples from a subsequent rat study were analysed for their true gastric total protein digestion [53]. The presence of actinidin here improved the gastric digestibility of the PI. Meanwhile, actinidin had a positive effect on the digestibility of zein but had virtually no effect on the digestibility of wheat gluten. These studies highlight the relevance of the type of assay and measure of digestion to assess the effect of processing or modification on plant protein digestibility. 

Green lentil PI from conventional aqueous fractionation was hydrolysed with acid protease, actinidin, bromelain and papain, prior to in vitro digestion [54]. Intact proteins proved to be better hydrolysable than the protein hydrolysates. Nevertheless, as a net result, more low molecular weight peptides were produced from the protein hydrolysates than from intact PI. 

Hydrolysis positively affected the digestibility of rapeseed and rice bran PI. Fibre and protein from a rapeseed PI that was obtained by membrane processing were hydrolysed [55]. The true digestibility of the hydrolysate was higher than the intact PI, as shown by a rat assay. As a result, the PDCAAS of the hydrolysate was also higher, compared to the original ingredient. Similarly, for progressively higher degrees of hydrolysis, a papain–hydrolysate of rice bran PC was more extensively digested than the intact PI by pepsin–pancreatin digestion [56].

Chickpea protein hydrolysis did not alter the digestion. Neither alcalase, flavourzyme [57], trypsin, papain nor pepsin [58] changed the extent of protein digestibility in in vitro assays. Nevertheless, the TIA was significantly reduced by the hydrolysis [57].

### 4.2. Heat Treatment

The process step most studied in terms of its effect on protein digestibility is heat treatment. Different conditions as well as different crops have been studied with positive, neutral or negative effects of heating on protein digestibility.

It is commonly thought that a certain extent of heat induced protein denaturation improves the digestibility, while more extensive heat treatment would induce protein aggregation which would, in turn, reduce the digestibility. The work of Tian et al. [59] demonstrates the relation between heating time and temperature, and the extent of pepsin-catalysed hydrolysis of soybean PI. Dispersions heated at 85 °C for 15 min presented the highest DH, while those heated at 70 or 100 °C were hydrolysed to a significantly lesser extent. In terms of time, PI heated at 85 °C for 20 min showed the highest DH compared to those heated for 10 or 60 min. Overall, all heated samples were more extensively hydrolysed than the unheated control. 

Soybean is one of the crops most widely studied in terms of the effect of heat treatment on protein digestibility. Studies have shown improvement but also reduction of protein digestibility as a result of heat treatment. The in vitro pepsin–pancreatin digestibility of soybean PI was improved by relatively short heating for 15 min at 95 to 121 °C [60,61,62]. β-conglycinin is known to be less susceptible to pepsin-catalysed hydrolysis than glycinin. Nevertheless, the gastric digestibility of both fractions was improved by heat treatment [60]. In this study, heating induced protein aggregation as well as pepsin during the gastric phase. The TIA of germinated soybean PI was reduced by the heat treatment [25]. Conversely, the apparent digestibility of heated, spray-dried and autoclaved pastes of soybean PI, determined in a rat assay, was significantly lower than that of non-autoclaved pastes [63]. Besides the heat treatment during drying, these pastes were autoclaved for up to four hours, highlighting that extensive heat treatment, both in time and temperature, has a detrimental effect on protein digestibility.

Heat treatment does not affect the digestibility of different pulse protein ingredients in the same way. Heating at 95 °C for 30 min improved the pepsin–trypsin digestibility of mung bean PI, reduced it for red bean PI, and did not change it for red kidney bean PI [64]. A larger extent of aggregation in heated mung bean PI was reported than in red kidney bean PI. It was suggested that the presence of basic, hydrophobic and uncharged polar AA influences the thermal and structural stability of proteins, and thus the tendency to aggregate when heated. Meanwhile, the in vitro digestibility of lupin and winged bean PC was improved by heating in a boiling water bath for up to 30 min [65,66]. The trypsin and chymotrypsin inhibitory activity of the freeze-dried winged bean PC was inactivated by heat treatment [66].

Likewise, the digestibilities of individual protein fractions from different crops are not modified in a similar manner upon heating. Vicilin-like proteins from chickpea and common bean are both resistant to gastric digestion; however, the digestibility of the former was improved by autoclaving, while for the latter, it was reduced [67,68]. Furthermore, chickpea albumin, 11S and total globulin digestibility increased, as a result of heat treatment [68]. Conversely, native protein fractions from fava bean were better digestible than those that denatured after autoclaving [67]. 

One might expect that preventing heat-induced aggregation would lead to a positive effect on protein digestibility. This was observed for lentil globulins which were unsusceptible to heat-induced aggregation, given that disulphide interactions were not observed [69]. Nevertheless, the negative charge of a protein fraction from common bean made the protein less prone to aggregation and yet less digestible than its unheated, less negatively charged, counterpart [67]. Based on the effect of heating on the electric charge of proteins and peptides, the latter study suggested that protein electronegativity and hydrophobicity were associated with protein aggregation and digestibility.

Net-zero effects may result from concurring events improving and reducing the DH achieved by digestive enzymes. Commercial soybean and pea PI dispersions heated at 90 and 120 °C for 30 min did not show different DH during in vitro gastric digestion compared to their unheated counterparts. Upon close inspection of the soluble and sedimented tailings, we found that heating improved the solubility of the commercial PI, and that the proteins separated into this fraction could be more extensively hydrolysed than those in the sedimented fractions [70].

Meanwhile, for dry-fractionated ingredients, heat treatment has shown to reduce the gastric digestibility of lupin and quinoa proteins. More small peptides (<3 kDa) were released from the unheated and heated at 60 °C dispersions of air-classified lupin PC than the dispersion heated at 90 °C [71]. A similar trend was observed for dry fractionated quinoa PC, with unheated and heated at 60 °C dispersions being more extensively hydrolysed than dispersions heated at 90 and 120 °C [45,72]. Similarly, quinoa PI from conventional aqueous fractionation showed lower DH with increasingly higher heating temperatures [30].

As previously discussed, alkaline heat treatment is generally detrimental for protein digestibility. Heating at higher pH reduced the in vitro protein digestibility of globulins from navy bean [73], of soybean PI [74] and rapeseed PC [75]. These results were confirmed for spray-dried soybean PI by an in vivo rat study [76]. For the most part, the limited digestibility can be attributed to the formation of lysinoalanine at high pH [74,75,76]. 

Thus far, we discussed studies on so-called moist heating, but the environment during heating does influence the protein digestibility. Sathe, Iyer and Salunkhe [14] compared dry and moist heating of navy bean PC and PI extracted with Na_2_CO_3_, as well as water-extracted albumins and NaCl-extracted globulins. The DH achieved with trypsin-α-chymotrypsin-peptidase was improved more significantly by moist than by dry heating. Similarly, boiling, microwaving, autoclaving, and dry or oven heating improved the digestibility of sweet potato PC [77]. Autoclaved dispersions presented the highest DH by pepsin–pancreatin digestion, followed by microwave and, lastly, dry heating. The PDCAAS determined in a rat assay was improved for autoclaved PI compared to the unheated ingredient. As previously reported, the TIA was reduced by all types of heat treatments studied.

These observations give a sense of the optimum range of heat treatment to improve the protein digestibility; more heating can negatively impact the digestibility (Table 1). The appropriate heat treatment would then depend on the ingredient source, the type of protein fraction, the type and conditions of heating.

### 4.3. High Pressure Processing

Laguna et al. [78] conducted a comprehensive study on the effect of heating and high pressure processing at two different pH (3.6 and 6.2) of commercial pea PI on its in vitro digestibility. For the most part, high pressure processing improved the gastric digestibility of pea protein. Samples prepared at a higher pH were more digestible than those at pH 3.6. Autoclaving did not alter the protein digestibility at either pH, which shows that the effect of pressure cannot be explained by denaturation, similar to that during heating. High pressure processing followed by a 30 min, 80 °C heat treatment at pH 3.6 reduced the protein digestibility. In contrast, high pressure processed red kidney bean PI presented a significantly lower in vitro digestibility by trypsin [79]. This was attributed to the generally low digestibility of phaseolin, particularly when aggregated. In this case, we may conclude that the protein source, as much as the processing steps, influences the digestibility of proteins.

## 5. Crosslinking, Complexation and Other Modifications

Forming protein complexes with other proteins or other compounds can be an unintended consequence of combining materials in one matrix or can be intentionally induced to achieve certain functions, such as colon-targeted drug delivery [80] or to confer an added nutritional benefit [81].

### 5.1. Transglutaminase-Catalysed Polymerization

Phaseolin from *Phaseolus vulgaris* L. was cross-linked by microbial transglutaminase [80]. Its isopeptide bonds made phaseolin more resistant to pepsin and trypsin action, especially for pepsin. Similarly, the pepsin–trypsin digestibility was reduced for native and heated crosslinked proteins from soybean PI, while it was improved by heat treatment alone [82]. While a single protein source was used in this study, covalent crosslinks were identified between β-conglycinin and acidic subunits of glycinin. In contrast, positive effects on the trypsin digestibility as a consequence of crosslinking by transglutaminase have been reported in red kidney bean PI [83]. The digestibility increased in crosslinked protein with longer crosslink reaction times, which was attributed to protein unfolding and denaturation of the vicilin unit.

Limited protein degradation by pepsin–pancreatin was observed for soybean PI polymers and heteropolymers with whey PI or casein, compared to the untreated PI [84]. Furthermore, soybean PI heteropolymers were more resistant to in vitro digestion compared to the whey PI-casein heteropolymer. This was attributed to reduced accessibility for enzymes to the peptide bonds, due to blockage of lysine residues and steric hinderance. Likewise, soybean PI–bovine gelatine composites showed lower pepsin–trypsin digestibility than the PI [85]. Trypsin-catalysed hydrolysis, prior to in vitro digestion, increased the digestibility slightly but it remained significantly lower for the untreated PI.

Glycation and crosslinking soybean PI with chitosan, or oligo-chitosan with transglutaminase improved the pepsin–trypsin digestibility [86,87]. The crosslinked soybean PI was more digestible than the untreated PI in both pepsin and pepsin–trypsin digestion assays. 

To assess the effect of Maillard reaction products, crosslinked commercial soybean PI was heated with D-ribose or sucrose [88]. Crosslinking had a negative effect on in vitro protein digestibility, particularly at longer transglutaminase incubation times. Overall, sucrose-containing samples were more digestible than ribose-containing samples. AA loss was reported as a consequence of crosslinking, most significantly of lysine.

Therefore, the effect of transglutaminase-catalysed crosslinking on protein digestibility depends on the extent to which cleavage sites become exposed or buried within the structure of the crosslinked protein. Furthermore, AA bioavailability could also decrease as a result of this processing step.

### 5.2. Acylation

Acylation of proteins can result in techno-functionality, such as solubility and emulsifying activity [89]. Mung bean PI was acylated with succinic and acetic anhydrides [90]. The trypsin–pancreatin digestibility was improved by acylation, probably due to protein unfolding. Acetylation was reported to reduce antinutritional factors (phytic acid, tannins and trypsin inhibitors) to a greater extent than succinylation. Similarly, acetylated and succinylated red kidney bean PI were more digestible by trypsin than their untreated counterpart [89]. This was attributed to increased protein solubility and protein unfolding. 

The improved digestibility due to acylation observed with these ingredients was also reported for a soybean PI hydrolysate [91]. The in vitro digestibility was significantly higher for succinylated soybean PI hydrolysates compared to the non-succinylated control. The authors also attributed this effect to protein dissociation or unfolding, and an increase in solubility. de Regil and Calderón de la Barca [92] assessed the in vivo digestibility of a soybean protein hydrolysate enzymatically bound by chymotrypsin to methionine methyl-ester using a rat study. There was no significant difference between the apparent digestibility of modified soybean PI hydrolysate and the control with free methionine. Nevertheless, the protein efficiency ratio was significantly higher for the modified ingredient.

Again, protein unfolding is related to an improvement of its digestibility, as was also observed with thermal denaturation. Moreover, peptides of lower molecular weight and, perhaps as a result, increased solubility would generally result in better digestibility, unless opposed by other cross-effects. 

### 5.3. Complexation with Phenolic Compounds

The digestibility of thermally denatured soybean PI was significantly improved, mostly by pepsin, when complexed with anthocyanins from black rice extract [62]. It was suggested that the network formed by the complex promotes enzymatic action is made possible by changes in the secondary structure; again, (partial) unfolding then facilitates the digestion. In a similar manner, soybean PI–curcumin complexes were more extensively hydrolysed than the non-complexed PI, particularly by pepsin, in a sequential pepsin–pancreatin in vitro digestion assay [60]. Heating before complexation did not influence the extent of digestion of the proteins. Furthermore, the typically pepsin-resistant β-conglycinin unit was completely degraded when it was part of the curcumin nanocomplex. Budryn et al. [93] studied soybean PI–hydroxycinnamic acids complexes, either individual 5-caffeoylquinic acid, caffeic acid or ferulic acid, combined in green coffee extract or encapsulated in β-cyclodextrin. The reduction in average molecular weight after pepsin-(trypsin-chymotrypsin) digestion was greater for the complexes than for the untreated PI. It was suggested that interactions and exposure of hydrophobic AA were responsible for the enhanced digestibility, although proteases might also interact directly with hydroxycinnamic acids. 

In contrast to the positive effects of anthocyanins and hydroxycinnamic acids, protein–polyphenol complexes reduce the digestibility of pea and soybean PI. Nine commercial pea PI with different physical and chemical characteristics were used to form complexes with polyphenols from cranberry pomace [81]. For some PI, no significant differences were found in the pepsin digestion of non-complexed and complexed proteins; however, all complexed isolates were less extensively hydrolysed by pancreatin digestion. The digestion rate was inversely related to the particle size of the PI. Similarly, soybean PI complexed at 70 or 121 °C with polyphenols and flavonoids from black soybean seed coat extract, was less extensively hydrolysed by pepsin–trypsin than the non-complexed ingredient [61]. Moreover, the DH was further reduced by increasing extract concentrations used to produce the complexes. Extract–enzyme or extract–protein interactions were thought to alter the digestive enzymes’ conformation, rendering them inactive for protein hydrolysis. In a rat assay, the true nitrogen digestibility was reduced for soybean PI that was complexed with both chlorogenic acid and quercetin [94]. The PDCAAS was significantly reduced for derivatized protein with lysine being the limiting AA.

Yang et al. [95] proposed a multistep process to produce a fermented soybean milk enriched with isoflavone aglycone. More intact proteins remained after pepsin–trypsin hydrolysis of the soybean PI–isoflavone complex, than of the PI. The isoflavone probably inhibited the protease activity. Nevertheless, heated and fermented soybean PI–isoflavone were more extensively hydrolysed than their unheated or non-fermented counterparts.

Phenolic compounds can modify the conformation not only of the proteins but also of the digestive enzymes. Changes in protein conformation can have a positive or negative effect on protein digestion. The former, if unfolding leads to the exposure of cleavage sites, or the latter, if it leads to steric hinderance surrounding the cleavage sites. Furthermore, phenolic compound could also act as inhibitors when bound to the digestive enzymes.

### 5.4. Protein Oxidation

Zhao et al. [96] found that a certain extent of protein oxidation had a positive effect on the soybean protein gastric digestibility as a result of protein unfolding, particularly for glycinin. However, severe treatments, i.e., by lipoxygenase-catalysed linoleic acid oxidation [96] or by incubation with 2,2′-azobis (2-amidinopropane) dihydrochloride [97], had a negative impact. In the latter study, the action of the radical-generating compound did not affect the gastric digestion, but it reduced the DH by pancreatin in the intestinal phase. This effect was directly influenced by increasing concentrations of the compound in the system. It was shown that oxidation can degrade several AA and induce protein aggregation. Sánchez-Vioque et al. [98] attributed a reduction in digestibility of chickpea legumin mixed with linolenic acid, to protein oxidation or non-covalent protein–lipid interactions. Meanwhile, no clear relation between carbonyl content, from oxidation products, and extent of hydrolysis in the gastric phase has been observed in thermomechanical processed soybean PC and PI [99].

### 5.5. Other Modifications

Soybean PI incubated with malonaldehyde, a lipid peroxidation product, was subjected to in vitro pepsin–pancreatin digestion [100]. β subunits of β-conglycinin were somewhat degraded by pepsin but they became more resistant to pancreatin digestion with increasing malonaldehyde concentration. The availability of indispensable and total free AA after digestion decreased in modified soybean PI.

Soybean PI, cottonseed PC and peanut PC formed complexes with glucose or sucrose [101]. In vitro digestibility was reduced by longer heating times to form the complexes. Protein–glucose complexes were less digestible than the sucrose complexes. Further, available lysine was reduced with heat treatment. 

Lastly, soybean PI was incubated with phytase from *Aspergillus niger* to obtain ingredients with different phytate contents [102]. Phytate content, parallel to TIA, was inversely related to pepsin–pancreatin digestibility.

Repeatedly, we find that any process or modification that would induce a certain degree of unfolding will generally facilitate digestion, but extensive unfolding leading to aggregation will result in slower or reduced digestion. Furthermore, modification of AA, particularly of lysine, will often lead to their reduced bioavailability. Finally, processes that reduce or inactivate antinutritional factors, such as phytate or protease inhibitors, will also improve or facilitate the digestion of proteins.

## 6. Structure Formation

### 6.1. Extrusion and Texturization

The in vitro digestibility of yellow pea and soybean PC can be improved by extrusion. The barrel temperature and screw speed are positively related to the protein digestibility of air-classified pea PC, while the moisture content has a negative influence on its digestibility [103]. Soybean PC, maize meal and cassava root starch were mixed and extruded [104]. The samples extruded at the highest temperature, moisture content and screw speed were the most digestible. The TIA, phytic acid and cyanide contents were reduced by extrusion; however, the tannin content was not reduced. Higher temperatures during extrusion led to more digestible proteins, which opposes the observations from moist heat treatments (Section 4.2), the reason is not fully understood and requires further research.

Duque-Estrada, Berton-Carabin, Nieuwkoop, Dekkers, Janssen and van der Goot [99] explored the effect on in vitro gastric digestibility of high temperature shearing of soybean protein ingredients, as well as the relevance of structure and size reduction in the digestibility. Sheared samples were cut into small pieces or ground into finer particles. Pepsin-catalysed hydrolysis was faster for unheated dispersions, followed by ground matrices. Cut samples were more slowly and less extensively hydrolysed than the other physical states.

The work from Li et al. [105] shows how the formation of rice glutelin fibrils through heat treatment under acidic conditions makes the protein more resistant to pepsin–pancreatin digestion. 

### 6.2. Pre- and Intra-Gastric Gelation

Opazo-Navarrete et al. [106] related the mechanical strength and porosity of heat-induced gels of soybean PI and pea PC to their gastric digestibility. No significant differences were observed between gels pre-heated at different temperatures. Soybean protein gels were less extensively hydrolysed than the control consisting of a protein dispersion, unlike pea protein gels that were hydrolysed to a similar extent as the dispersion.

Pressure-induced gels from air-classified lentil and fava bean PC were more digestible than heat-induced gels under in vitro gastric conditions [107]. It was suggested that the network of pressure-induced gels allowed for a similar extent of access to pepsin as in concentrated protein dispersions. Meanwhile, both treatments changed the structure of the 55 kDa fractions to be better digestible in the gastric phase. The TIA was more significantly reduced by heating than by pressurization. 

Soybean PI coagulates formed with MgCl_2_ or glucono-δ-lactone were more digestible than gels prepared with transglutaminase [108]. This was attributed to the covalent iso-peptide bonds formed by transglutaminase that cannot be degraded during in vitro gastrointestinal digestion. In contrast, the non-covalent bonds formed during coagulation by MgCl_2_ or glucono-δ-lactone could be broken during digestion. Soybean PI and glycerol films were prepared with ferulic acid, tannin, corn starch or H_2_O_2_ at pH 7 to 10 [109]. The gastric digestibility of the films was significantly lower than that of the PI in a dispersion, except for the films prepared with corn starch, which were digested to a similar extent as the control. Lysine availability was also lower in the films. Ferulic acid and tannins were thought to form crosslinks with AA, while H_2_O_2_ could have oxidized certain AA. Lastly, films formed at pH 9 and 10 were less digestible than at pH < 8.5. This was attributed to AA isomerisation and crosslinking at high pH. 

In a simulated gastric environment, dispersions of soybean PI and negatively charged polysaccharides (xanthan gum, carrageenan [110] or alginate [111]) self-assemble into a hydrogel. The pepsin-catalysed hydrolysis of the gels was slower even at low polysaccharide contents, compared to the single PI. Hu, Chen, Cai, Fan, Wilde, Rong and Zeng [110] similarly found that soybean PI–carrageenan gels were digested more slowly than those with xanthan gum, due to the more compact and dense gel network in the former.

Generally, structure formation led to a slower and sometimes lower extent of hydrolysis by digestive enzymes compared to liquid dispersions. This is explained by physical hinderance surrounding the cleavage sites. Therefore, looser structures as weaker gels allow for a better digestibility than tighter structures. Further, covalent crosslinking inhibits protein unfolding, while non-covalent bonds can dissociate, especially at lower pH in the stomach, and thus allow for faster digestion. As heat treatment is often required before gelation, antinutritional factors, such as trypsin inhibitors, can also be inactivated.

## 7. Macronutrient Interactions

Proteins are almost never processed or consumed on their own. The effect of the interaction of proteins with other macronutrients on protein digestion is not fully understood, but there are some general directions suggested.

### 7.1. Animal- and Plant-Based Protein Hybrid Foods

Reconstituted beverages containing the combination of bovine milk PC and soybean, pea or rice PI showed an improved in vitro DH and PDCAAS of blends compared to individual plant proteins [112]. However, this was not observed in solid matrices. Proteins from pea PI, rice protein or lentil flour were enzymatically bound to beef chuck ground meat using transglutaminase [113]. The cooked restructured beef steaks were digested using the INFOGEST 2.0 model with expectorated boluses. No outstanding differences were observed in the peptide size distribution in the digestates of the samples with different treatments. Lentil-enriched steaks released the highest amounts of free isoleucine, lysine, phenylalanine and valine. Protein (re-)aggregation was observed after in vitro gastrointestinal digestion. 

### 7.2. Starch

Oñate Narciso and Brennan [114] found a relationship between the amylose content of starch with protein digestion. Pea PI was combined with starch from basmati and glutinous rice, with high and low amylose to amylopectin ratios, respectively. All proteins from the samples prepared with glutinous rice starch were degraded after pepsin–pancreatin digestion, but the vicilin and legumin acidic subunit from basmati rice starch samples remained after digestion. The authors proposed that the proteins were embedded into the amylose network. Similarly, quinoa protein from aqueous or dry fractionation was combined with starch-rich fractions from dry or mild aqueous fractionation, which after heating showed lower DH from in vitro gastric digestion than starch-free, unheated protein dispersions [45,72]. This reduction directly related to the heating temperature and was thus probably associated to starch gelatinization. Therefore, embedding the protein in a gelatinized starch gel does reduce the digestibility, probably due to the inaccessibility of the gel for the enzymes.

### 7.3. Fibre

The DH obtained by pepsin digestion of dry-fractionated quinoa PC was slightly reduced in quinoa fibre-containing unheated and heated dispersions [72]. The effect of fibre on quinoa protein gastric digestion was not as significant as for starch. Fibre seemed to counter the low hydrolysis induced by starch gelatinization. The fibre does not form a gel that is difficult to penetrate for enzymes but may induce somewhat better mixing due to the higher viscosity.

## 8. Conclusions

Plant proteins have the potential to provide all indispensable amino acids. However, as described at length, processing and plant protein digestibility are strongly related.

Heating and soybean are the process step and crop most researched, respectively, reflective of their ubiquity in the production of plant-based food products. Moderate heating may enhance the digestibility by inducing partial unfolding of the proteins, thereby rendering them better accessible for the proteases. However, extensive heating induces aggregation, which makes the cleavage sites less accessible. Similar effects are seen with other types of treatments. Acylation of protein-rich ingredients improved their digestibility, probably also due to partial unfolding. Meanwhile, alkaline treatment, during or after fractionation, consistently reduces the digestibility of different crops, since it strongly changes the structure of the protein and induces AA isomerisation. Again, we see an optimum in the severity of the treatments for digestibility. It is however clear that the exact impact depends on the origins of the proteins.

Ultimately, it is desirable to attain an overarching relationship between the digestibility and the modifications resulting from processing. This review can serve as a guide when considering a certain processing step in the production of plant-based alternatives to animal-sourced products. There are ample opportunities for further research of unexplored processes for promising crops and vice versa, to truly consider the use of plant protein-rich ingredients in food products as a transition pathway to a more sustainable food system.

## Figures and Tables

**Figure 1 foods-11-00870-f001:**
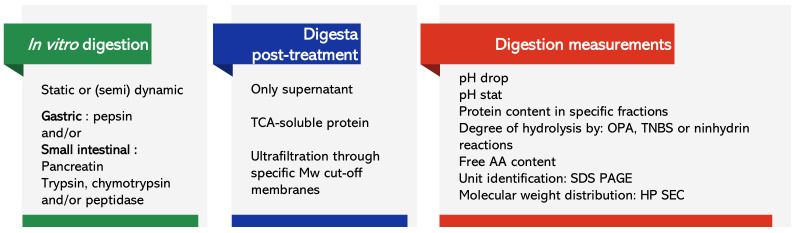
Characteristics of in vitro assays, treatment of digesta and description or quantification of *digestibility* in the studies reviewed. AA, amino acid; HP SEC, high performance size exclusion chromatography; Mw, molecular weight; OPA, *o*-phthalaldehyde; SDS PAGE, sodium dodecyl sulphate–polyacrylamide gel electrophoresis; TCA, trichloroacetic acid; TNBS, trinitro-benzene-sulfonic acid.

**Figure 2 foods-11-00870-f002:**
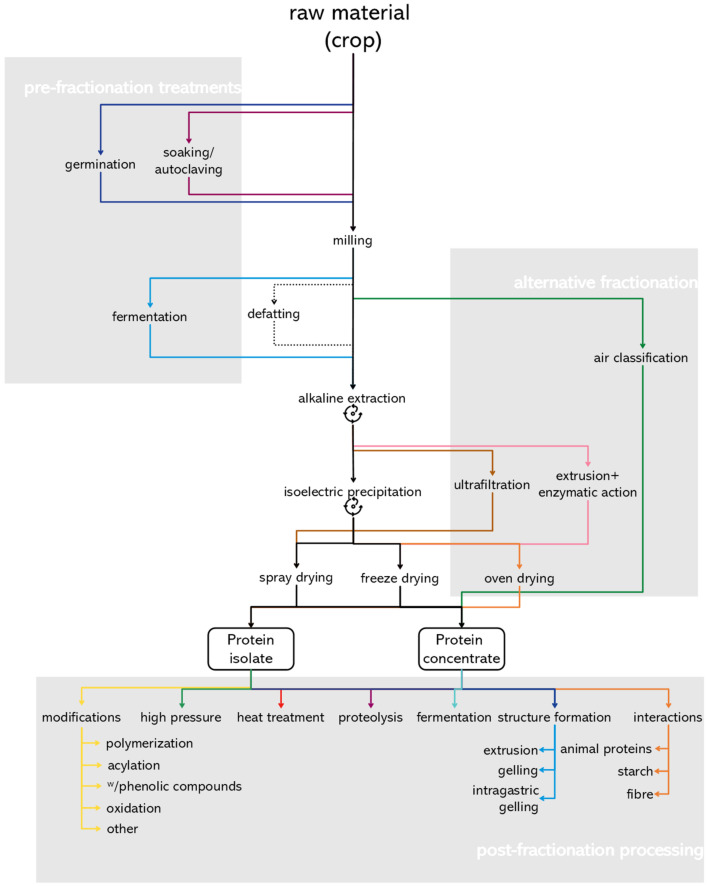
Overview of the processing steps before, during and after fractionation of plant proteins from the studies included in this review. Colours indicate the different routes for processing, the conventional route for aqueous fractionation is presented in black, and 
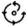
 represents centrifugation after alkaline extraction and isoelectric precipitation.

**Table 1 foods-11-00870-t001:** Overview of the effect of different types of processing before, during or after protein fractionation from different crops. 
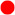
, 
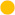
 negative; 
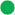
, 
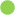
 positive; or 
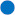
 neutral effect on protein digestibility. Only processes or ingredients with more than one study reporting on the effect of processing on digestibility were included in this table.

Crop	Pre-Fractionation	Fractionation	Processing	Nutrient Interactions Starch
Germination	Dry Fractionation	Alkaline Treatment	Fermentation	Enzymatic Hydrolysis	Heating	High Pressure	Polymerization	Acylation	Phenolic Compounds	Oxidation	Gelling	Extrusion
soybean	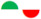	-	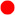 also post-fractionation	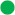	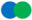	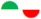	-	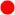	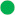 hydrolysate	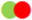	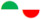	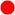	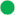	-
black bean	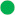	-	-	-	-	-	-	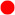	-	-	-	-	-	-
chickpea	-	-	-	-	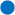	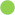	-	-	-	-	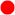	-	-	-
fava bean	-	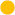	-	-	-	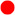 legumin	-	-	-	-	-	-	-	-
lentil	-	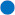	-	-	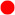	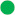 globulin	-	-	-	-	-	-	-	-
lupin	-	-	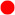	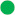 pre-fractionation	-	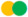	-	-	-	-	-	-	-	-
maize	-	-	-	-	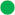	-	-	-	-	-	-	-	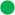	-
mung bean	-	-	-	-	-	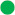	-	-	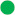	-	-	-	-	-
navy bean	-	-	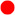	-	-	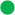	-	-	-	-	-	-	-	
quinoa	-	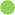	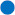	-	-	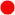	-	-	-	-	-	-	-	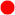
rapeseed	-	-	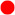	-	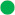	-	-	-	-	-	-	-	-	-
red kidney bean	-	-	-	-	-	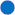	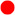	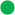	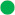	-	-	-	-	-
rice	-	-	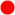	-	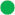	-	-	-	-	-	-	-	-	-
yellow pea	-	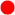	-	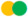	-	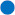	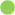	-	-	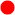	-	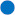	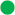	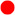

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
