# Peer review of "Effect of Fractionation and Processing Conditions on the Digestibility of Plant Proteins as Food Ingredients"

_foods, 2022, doi:10.3390/foods11060870_

Round 1
Reviewer 1 Report
The reviewed manuscript under title “Effect of fractionation and processing conditions on the digestibility of plant proteins as food ingredients” is focused on very important topic.
Author Response
We kindly appreciate the comments and input from the reviewers.
Reviewer 2 Report
The review focuses on the effects of fractionation and processing conditions on the digestibility of plant proteins as food ingredients. the manuscript is well-written and interesting.
Some comments about the manuscript:
- Change "plant proteins" to "plant-based proteins".
- Line 8: What do you mean by "plant protein-rich ingredients"? Any examples?
- Line 16-17: provide examples of the other nutrients and sources.
- Table 1: There are recent studies on the effects of fermentation and pH recycling on lentil proteins.
- Figure 1: Why start with "seeds"? Try to find another suitable term.
- Section 3.2: Insufficient info on conventional protein fraction.
- Section 4: Why is ultrasound-assisted processing not included?
Author Response
We kindly appreciate the comments and input from the reviewer.
- If the reviewer agrees, we would prefer to use the term ‘plant proteins’ to concur with the title of the special issue we have submitted to: ‘Functionality and Food Applications of Plant Proteins’.
- In the abstract, ‘plant protein-rich ingredients’ was substituted by ‘plant protein concentrates and isolates’ to specify.
- The abstract has been modified to include examples of the other nutrients (starch or fibre) and of the different sources. Details from other sentences were deleted to accommodate the word limit of the abstract.
-
One of the recent studies we found was:
Alrosan, M., Tan, T. C., Easa, A. M., Gammoh, S., Kubow, S., & Alu'datt, M. H. (2021). Mechanisms of molecular and structural interactions between lentil and quinoa proteins in aqueous solutions induced by pH recycling. International Journal of Food Science & Technology.
Unfortunately, we could not find any study relating fermentation and pH recycling with protein digestibility.
- The word ‘seeds’ was replaced by ‘raw material (crop)’.
- In section 2, the method for conventional aqueous fractionation was described. We find this section the most suitable for this information as this how protein isolates or concentrates were obtained in the post-fractionation processing studies.
-
The only study that was found relating ultrasound treatment (post-fractionation) with protein digestion was:
Martínez-Velasco, A., Lobato-Calleros, C., Hernández-Rodríguez, B. E., Román-Guerrero, A., Alvarez-Ramirez, J., & Vernon-Carter, E. J. (2018). High intensity ultrasound treatment of faba bean (Vicia faba L.) protein: Effect on surface properties, foaming ability and structural changes. Ultrasonics sonochemistry, 44, 97-105.
This study was mentioned in the first paragraph of Section 4.
In section 3.3, the study from:
Tang, C. H. (2007). Functional properties and in vitro digestibility of buckwheat protein products: Influence of processing. Journal of Food Engineering, 82(4), 568-576 also considered ultrasound treatment. This, however, during fractionation.
Reviewer 3 Report
This is a good well-written manuscript with an interesting title about the plant proteins which can be very useful for the scientists from different fields such as food science. This manuscript needs some modifications.
- I think the keywords are not selected well. Please use words that are not in the title.
-In my opinion, the importance of plant proteins for vegetarians and its trends should be mentioned in the introduction part of the manuscript.
- In the introduction, some limitations of plant proteins compared to animal proteins such as lower solubility and poorer functional properties and the processes used to improve these properties should also be mentioned.
-Recently, some studies have been done on the effect of fibrillation process on the properties of the plant proteins, but the authors did not mention this process in their manuscript.
As a drawback, this manuscript is more like a list of references and there are only few hypotheses from the authors themselves. They should add their own conclusion and suggestions about the observed differences between different studies which were used to prepare the various parts of this review manuscript.
Author Response
We kindly appreciate the comments and input from the reviewer.
- Keywords have been modified
- The sentence “With the increase of flexitarian diets, there is a growing demand in the vegetarian and vegan food space (Euromonitor International, 2020)” has been included.
- The limitation of plant proteins, compared to milk proteins, has been included in the second paragraph of the introduction.
-
Several studies were found on the fibrillation of β-lactoglobulin or ovotransferrin, as well as some plant proteins, such as soybean or kidney bean. Nevertheless, for the later the effect on protein digestibility has been seldomly researched. The work from: Lassé, M., Ulluwishewa, D., Healy, J., Thompson, D., Miller, A., Roy, N., ... & Gerrard, J. A. (2016). Evaluation of protease resistance and toxicity of amyloid-like food fibrils from whey, soy, kidney bean, and egg white. Food Chemistry, 192, 491-498 does study the in vitro digestion of the fibrils formed. However, in this work there was no control chosen, and it is therefore not suitable for this review.
In Section 6.1, we have included the work by: Li, S., Jiang, Z., Wang, F., Wu, J., Liu, Y., & Li, X. (2020). Characterization of rice glutelin fibrils and their effect on in vitro rice starch digestibility. Food Hydrocolloids, 106, 105918. -
We recognize and agree with the comment from reviewer 3. As can be perceived in the manuscript, the research relating processing and plant protein digestibility is quite sparse. Whenever several coinciding and contradicting studies, for instance for processing steps like heating and acylation, some overarching hypothesis and conclusions were drawn. For other processing steps only a few, and sometimes only one, studies were found, making it difficult to draw general conclusions.
In the conclusions, we touch upon this by mentioning the opportunities or suggestions for future research.